# Staircase Attention for Recurrent Processing of Sequences

**Da Ju**
Meta AI
daju@fb.com

**Stephen Roller**
Meta AI
roller@fb.com

**Sainbayar Sukhbaatar**
Meta AI
sainbar@fb.com

**Jason Weston**
Meta AI
jase@meta.com

## Abstract

Attention mechanisms have become a standard tool for sequence modeling tasks, in particular by stacking self-attention layers over the entire input sequence as in the Transformer architecture. In this work we introduce a novel attention procedure called staircase attention that, unlike self-attention, operates across the sequence (in time) recurrently processing the input by adding another step of processing. A step in the staircase comprises of backward tokens (encoding the sequence so far seen) and forward tokens (ingesting a new part of the sequence). Thus our model can trade off performance and compute, by increasing the amount of recurrence through time and depth. Staircase attention is shown to be able to solve tasks that involve tracking that conventional Transformers cannot, due to this recurrence. Further, it is shown to provide improved modeling power for the same size model (number of parameters) compared to self-attentive Transformers on large language modeling and dialogue tasks, yielding significant perplexity gains.

## 1 Introduction

Early breakthrough work in neural language modeling considered a fixed context size of tokens that are embedded with a lookup table, followed by nonlinearities and a final softmax to produce a probability distribution for the next output token in a sequence [5]. Such models were replaced, pre-Transformer, with recurrent models such as RNNs and LSTMs [12, 16, 28] that were able to consider arbitrary context length via the ability to store state in their memory using recurrent steps through the data, in contrast to the fixed length constraint of earlier models. Moreover, the repeated application of the recurrent network across the sequence also made the models considerably deeper: a given representation is a function of a large number of nonlinearities due to previous state. This allows such models to track state [1], store long-term memories, and potentially solve highly nonlinear sequential tasks. Today, with the advent of attention-based models [2] and in particular Transformers [36], fixed length inputs that eschew recurrence are back as the norm, thanks mainly due to deep stacks of nonlinearities on those fixed inputs that are also well suited to modern hardware, leading the authors of Vaswani et al. [36] to claim that non-recurrent attention is "all you need." However, some of the advantages just mentioned of earlier models – tracking state, and solving highly nonlinear sequential tasks – have to some degree been lost [13].

In this work, we introduce a novel recurrent model that utilizes a novel attention procedure called staircase attention. We show that our new model, which utilize both sequence aligned recurrence (in time) and recurrence in depth can bring advantages to modern models, in particular in terms of lower language modeling perplexities given the same number of parameters, and for solving nonlinear state-tracking tasks. Staircase attention, like self-attention, processes tokens in parallel for speed, but unlike self-attention, operates across the sequence (in time) recurrently processing the input by

---

[1]keep track of an evolving state given sequence of changes to it

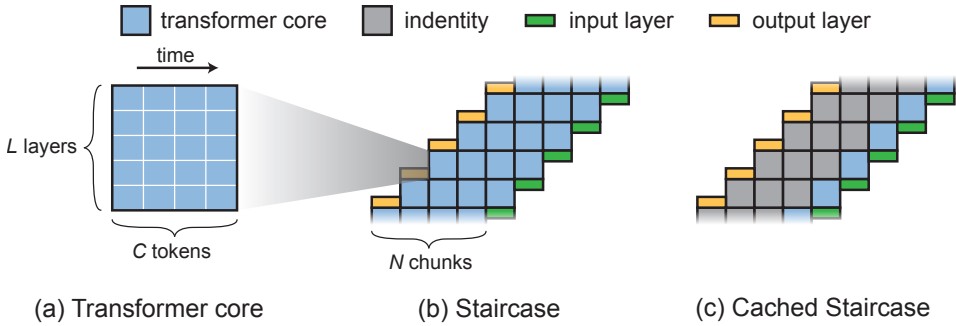

Figure 1: **Visualization of our Staircase models.** (a) The building block of our models is a Transformer core with $L$ layers that processes $C$ tokens (equals 1 chunk) in parallel. (b) In the *Staircase* model, Transformer cores are stacked diagonally, so each step sees one new input chunk. A column corresponds to processing of a single chunk and is composed of $N$ cores sandwiched between input and output layers. All $N$ chunks in a row are processed in parallel and are accessible by self-attention. (c) In the *Cached Staircase*, some final cores are replaced by an identity function to reduce computation. The output from the last core is cached and included within the attention span of later chunks, so the context size remains the same.

adding another step of processing. A step (processing block) in the staircase consists of backward tokens (encoding the sequence so far seen) and forward tokens (ingesting a new part of the sequence). Thus, on each time step the block moves forward in time, retaining a memory comprised of multiple vectors stored in the backward tokens (the recurrent tokens). The blocks utilize the same model weights for each step, hence giving a recurrence in depth.

Compared to Transformers, Staircase models can retain a recurrent memory in time, and repeated application of it in the recurrent network over the sequence also makes the model considerably deeper for the same number of parameters (but not necessarily the same amount of compute).

We show on two tasks requiring state-tracking that Staircase models can perform successfully where Transformers fail. We then show on two language modeling and a dialogue modeling task for the same number of parameters, significantly lower perplexities can be obtained compared to standard Transformers for certain kinds of Staircase models. We thus analyze our models and show that one can control the amount of recurrence and depth which can trade off compute for performance, which has practical implications depending on available memory and compute architecture constraints. The Staircase models perform well on both state-tracking tasks and language modeling tasks, providing good performance across the board. Our code will be made publicly available.

## 2 Related Work

Since attention was introduced [2], non-recurrent models have superseded in popularity recurrent models like RNNs and LSTMs [12, 16], which were for a time dominant in NLP applications, particularly when involving sequence generation. The first models to use stacked layers of attention over input word embeddings and position encodings, as a replacement to recurrence in time, were end-to-end memory networks [34]. Those models were shown on the task of language modeling to perform well compared to LSTMs, but in experiments still shared some weights across depth, which we refer to as *recurrence in depth* (also referred to as a "recurrent attention mechanism" in Vaswani et al. [36]). Transformers [36] removed the constraint of sharing any weights among layers at all, and showed this improves performance (at the cost of using more parameters). Transformers additionally contributed other notable improvements such as multi-head, self-attention and residual blocks. Such models do not have any recurrence at all, and are the current state-of-the-art architecture choice for many tasks.

Since then, several variants of Transformers have arisen that attempt to incorporate recurrence again by sharing some weights. The Universal Transformer proposes an extreme variant of applying the same layer (with shared weights) repeatedly [10]. Similarly, ALBERT [22] also shares the same weights across all layers for a pretraining/finetuning setting where the BERT model [11] is effectively

compressed; they also consider sharing only the self-attention or only the feed-forward weights. We note also that several works, while not sharing parameters of layers, have studied the ordering of the sublayers of Transformers, in particular Sandwich [29] and Macaron [25] Transformers.

Some works have also attempted to incorporate sequence-aligned recurrence to Transformers. Chen et al. [6] incorporate LSTM layers into the Transformer, and Hao et al. [15] blend a non-recurrent and recurrent model (e.g., an RNN) together with a gating function. Transformer-XL [9] employs a segment-level recurrence mechanism to effectively cache and speed up computations in long-context sequence tasks. We note that a number of recent architectures have also focused on allowing long-context in Transformers, although typically without employing recurrence [7, 21, 4]. Linearizing self-attention for efficiency [19, 31] uses state that is updated in a recurrent way, but the model still remains feedforward. Finally, the Feedback Transformer [13], perhaps the most similar work to ours, incorporates step-wise recurrence in the Transformer, with a step size of one and a fixed cached memory in the past. It achieves good results but has relatively high computational cost due to its architecture not fully exploiting parallelism.

In this work, we compare architectures with the number of model parameters fixed, and explore increasing recurrence and/or compute given that fixed budget. An orthogonal topic of study is to fix the compute budget instead, but do not fix the amount of parameters, e.g. research into large, sparse (typically non-recurrent) models that may require to be spread over a cluster [14, 23]. We focus on the former here, but learnings from each direction should be complementary.

## 3 Method

### 3.1 Background

In this paper, we consider decoder-only Transformers [1, 9] that are applied to sequential tasks like language modeling. In this setting, a Transformer model takes as input a sequence $\{x_1, x_2, \ldots, x_T\}$ of $T$ tokens and outputs a sequence of the same size

$$y_1, y_2, \ldots, y_T = \text{TRANSFORMER}(x_1, x_2, \ldots, x_T). \tag{1}$$

If we separate out the input embedding $\mathbf{h}_t = f_{\text{in}}(x_t)$ and the final output module $y_t = f_{\text{out}}(\bar{\mathbf{h}}_t)$, we are left with the Transformer core as shown in Figure 1a

$$\bar{\mathbf{h}}_1, \bar{\mathbf{h}}_2, \ldots, \bar{\mathbf{h}}_T = \text{TC}(\mathbf{h}_1, \mathbf{h}_2, \ldots, \mathbf{h}_T), \tag{2}$$

which consists of $L$ layers that compute final hidden states for each token. Each layer is composed of self-attention and feedforward sublayers. In the self-attention sublayer, causal masking is applied to prevent tokens from attending to any future token, and we use relative position embeddings [32]. See Vaswani et al. [36] for more details about the sublayer architecture of Transformers.

### 3.2 Staircase Model

We now describe our model that utilize staircase attention. Later, we also introduce *Cached Staircase* version that is more computationally efficient. Their graphical representation may be found in Figure 1.

Unlike Transformers, a Staircase model ingests input tokens in smaller chunks, as shown Figure 1b. Inside a Staircase model lies a Transformer core that processes each input token in $N > 1$ recurrent steps. With each recurrent step, a Staircase model moves $C$ tokens forward in time, which we call the *forward* step size. In addition to those $C$ forward tokens, the model simultaneously also processes $NC - C$ tokens that come from the previous step, which we call *backward* tokens. We refer to the total number of tokens $NC$ that are being processed in parallel as the *step size*.

Let us denote a chunk of C input tokens as $\mathcal{H}_i^0 = \{\mathbf{h}_{iC+1}, \ldots, \mathbf{h}_{iC+C}\}$ for brevity. Here $\mathbf{h}_t$ is the embedding vector of the token $x_t$. At each step, the Staircase model processes $N$ chunks in parallel via the Transformer core

$$\mathcal{H}_{i+1}^N, \mathcal{H}_{i+2}^{N-1}, \ldots, \mathcal{H}_{i+N}^1 = \text{TC}(\mathcal{H}_{i+1}^{N-1}, \mathcal{H}_{i+2}^{N-2}, \ldots, \mathcal{H}_{i+N}^0). \tag{3}$$

Among the input chunks, only $\mathcal{H}_{i+N}^0$ contains new token embeddings (i.e., forward tokens) while the remaining $N - 1$ chunks come from the previous recurrent step (i.e., backward tokens). The

superscript $n$ of $\mathcal{H}_i^n$ denotes the number of computational passes through the Transformer core. After $N$ passes through the core module, the output for a particular token $x_t$ is computed with

$$y_t = f_{\text{out}}(\bar{\mathbf{h}}_t) \quad \text{for all } \bar{\mathbf{h}}_t \in \mathcal{H}_i^N.$$

As you can see, an input token gets processed by the same core module $N$ times. This makes it possible to control the amount of computation by varying the number of recurrent steps $N$ without changing the number of parameters of the model.

Feeding states computed by the previous step into the next step computation makes Staircase models recurrent in time like RNNs because each recurrent step moves forward $C$ tokens. There are two advantages to this type of recurrence. First, the number of non-linear computations between an input token $x_t$ and output token $y_{t+\tau}$ linearly increases with their distance $\tau$. In contrast, Transformers are strictly a feedforward model that has a fixed number of computation steps. The second advantage is that information can pass to future steps without any limits, whereas standard Transformers are limited by their token truncation length. These two advantages make recurrent models capable of maintaining an internal state, but more importantly of updating this state continuously through time. However, unlike RNNs, Staircase models take advantage of the attention mechanism in the same way as Transformers, and store state across multiple vectors: the $NC - C$ backward tokens. Like Transformers, they thus take advantage of parallelism.

We use the same techniques as Transformer-XL [9] for processing very long or unbounded sequences. First, each token will attend to a fixed number of previous tokens $S$ rather than the whole sequence. This reduces the computational complexity of the self-attention from $\mathcal{O}(T^2)$ to $\mathcal{O}(TS)$ assuming $S \ll T$. Next, we split the input sequence into smaller manageable segments and let the model process them sequentially. To avoid the segment boundaries from obstructing attention across segments, the hidden states computed in the previous segments are fixed and kept in memory. Then, this fixed-memory is made available in the self-attention sublayer for subsequent segments so a token can attend to a token in the previous segment. See Dai et al. [9] for more details about this mechanism.

### 3.2.1 Cached Staircase Model

In Staircase models, the self-attention sublayer processes $NC$ tokens at a time. This means how far a token can directly attend to is limited by this context size $NC$. However, the hyperparameter $N$ also controls the number of recurrent computations, and one might want to decouple these two factors to control context size versus recurrence.

Here we propose a simple solution for increasing the context size while keeping the recurrent computation constant. We do this by introducing a new hyperparameter $M < N$ and put hidden states in a *cache* after $M$ recurrent steps. Figure 1c shows a case where $N = 4$ and $M = 1$. Once a hidden state is in the cache, it stays the same, requiring no additional computation

$$\mathcal{H}_i^n = \mathcal{H}_i^M \quad \text{for } n > M.$$

This means the number of recurrent computations on a particular input is limited to $M$. However, hidden states stay in the cache for the remaining $N - M$ steps so other tokens still can attend to them. This is achieved by including cached hidden states only when computing keys and values in the self-attention sublayer of the Transformer core. As a result, the self-attention sublayer will have $NC$ keys and values, but only $MC$ queries, reducing its computational complexity from $\mathcal{O}(N^2C^2)$ to $\mathcal{O}(NMC^2)$. As cached hidden states are excluded from the feedforward sublayer altogether, the computational complexity there changes from $\mathcal{O}(NC)$ to $\mathcal{O}(MC)$. Thus, the context size $NC$ can be increased by picking a larger $N$, but the amount of computation can be reduced by choosing a smaller $M$. For example, for $M = 1$, we can see that the reduction in computation is $N$ fold.

### 3.3 Relation to Existing Models

**Transformer** The standard Transformer corresponds to a Staircase model with a large chunk size and no recurrence. While it is efficient at processing tokens in parallel, it has no ability of retaining and recomputing state across sequences, other than by fitting those tokens into the current processing block.

Table 1: **Results summary across all our tasks.** We compare four architectures where we fix the number of learnable parameters to be the same for all models on the same task. * is from [18].

| Model | Random Walk (error %) | Algorithm (error %) | Enwik8 (test bpc) | Reddit (test ppl) | BASE Data (valid ppl) |
|---|---|---|---|---|---|
| LSTM | 0.6 | 1.0* | 1.38 | - | - |
| Transformer-XL [9] | 84.1 | 48.7 | 1.15 | 26.2 | 28.0 |
| Feedback Transformer [13] | **0.1** | **0.2** | **1.12** | 25.5 | 26.6 |
| *Our models* | | | | | |
| Staircase | **0.1** | **0.2** | 1.14 | **22.6** | **23.0** |
| Cached Staircase | **0.1** | 1.2 | 1.13 | 26.1 | 27.8 |

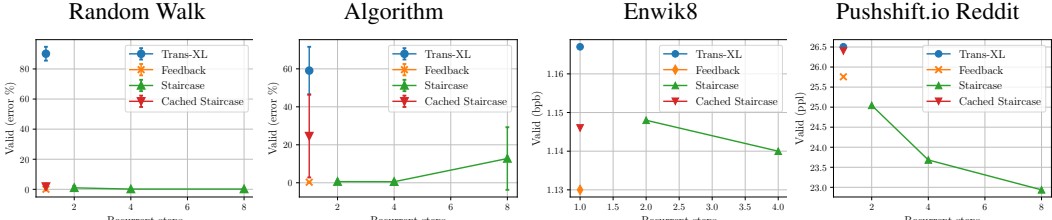

Figure 2: The performance of various models on the four tasks with different numbers of recurrent steps. The number of parameters is the same for all the models.

**Feedback Transformer**    The Feedback Transformer [13] is equivalent to a Cached Staircase model with a chunk size of $C = 1$ (i.e., a forward step of a single token), and caching after $M = 1$ step, i.e. all tokens in the past are part of a fixed cached memory. In contrast, larger chunk sizes allow general Staircase models to exploit parallelism and be far more efficient than the Feedback Transformer, while increasing $M$ can give more expressive power. We compare to this model in our experiments.

**Recurrent Neural Networks**    RNNs [12] that store recurrent state in a single vector and ingest tokens one at a time can be compared to a Staircase model with a single backward token and a single forward token, i.e. a chunk size of $C = 1$ and $N = 2$. Staircase models exploit parallelism similar to Transformers while maintaining several chunks of recurrent (per token) features to more expressively track state than conventional RNNs.

**Memory Networks**    MemNets as implemented in Sukhbaatar et al. [34] employ recurrence in the stacked layers of attention and computation for the current token, but only compute input embeddings $\mathbf{h}_t^0 = f_{\text{in}}(x_t)$ for previous tokens, and can thus be seen as a Cached Staircase with a chunk size of $C = 1$ and caching at all previous steps, $M = 0$.

**Transformer-XL**    Transformer-XL [9] uses sliding-window attention where each token attends to a fixed-number of previous tokens. Like Cached Staircase, it also has a caching mechanism which eases computation when dealing with earlier chunks of tokens. The Staircase model has a similar sliding attention window, but its unit is a chunk of C tokens instead of one. The more important difference is that Staircase models take the last state of earlier chunks and process that state further in a recurrent way; Transformer-XL on the other hand extends each layer of the Transformer's attention mechanism to using old cached states at each layer, i.e. does not build further computations on top of the old state. We use this as a baseline in our experiments.

**Universal Transformer**    Universal Transformers [10] propose to tie all the layer weights in a Transformer. In contrast Staircase models repeat multiple layers with different weights, thus allowing a large number of parameters without prohibitive computational cost.

# 4 Experiments

We use two types of tasks to test our models and compare its variations, along with Transformer-XL [9] and Feedback Transformer [13] baselines. First, we have two artificial state tracking tasks specifically designed to test the model's ability to keep track of evolving changes. Next, we use real-world language modeling tasks. See Appendix A for further details of our experimental setup for training, including all hyperparameter choices. We also make the staircase implementation public available on GitHub[2].

## 4.1 Tasks

**Random Walk**  At each time step, an agent in a small grid takes a random action that turns the agent, or moves it forward. A model has to predict the agent's location given these actions. This seemingly simple task requires recurrency and has been shown to make feedforward models like Transformers fail. We borrow this task from Fan et al. [13], but increase the length from 100 to 400 to make it more challenging. See Fan et al. [13] for more details about this task.

**Algorithm**  This task consists of simple algorithmic operations that need to be executed sequentially. Some operations depend on the current variable values, which makes it necessary to keep track of variable values and update them if necessary. Thus, it also requires recurrency, and like the Random Walk task has been shown to make Transformers fail. We use the 3 variable version of the task from Fan et al. [13].

**Enwik8**  Enwik8 is a character-level language modeling task [26] that consists of 100M tokens taken from Wikipedia articles. The challenging part of this data is that it requires long-term reasoning [35] because tokens are characters instead of words.

**Pushshift.io Reddit**  We use a variant of Reddit discussions, which has also been used in several existing studies [37, 27, 20, 33]. Following Humeau et al. [17], we use a previously existing Reddit dataset extracted and obtained by a third party and made available on pushshift.io [3], training to generate a comment conditioned on the full thread leading up to the comment, spanning 1.5B training examples from Reddit obtained from Pushshift through July 2019. See Roller et al. [30] for more details. We concatenate the dataset to view it as a language modeling task.

**BASE Data**  We use the language modeling dataset from Lewis et al. [23], which consists of approximately 100B tokens, combining the corpora used in Liu et al. [24] that consists of Wikipedia, BookCorpus, CC-News, OpenWebTex and Stories, along with the English subset of the CC100 corpus [8].

## 4.2 Results

Our results on all of the tasks are summarized in Table 1. For each task, all the models have the same number of parameters and use the same Transformer core architecture implementation. For Random Walk and Algorithm tasks, we trained each model with multiple seeds and chose the best seed as measured by their validation performance. The specific configuration of each model can be found in Table 5 in Appendix A. We see clear wins from our models on all the tasks, and the following subsections will analyze these results in detail.

### 4.2.1 Staircase models have strong performance on state tracking tasks

The Random Walk and Algorithm tasks are specifically designed to test a model's capability of tracking states: to store given information internally and update it with new information coming at each time step. In Table 1 we report results from running multiple training seeds, and selecting the one with best performance on the validation set. In Figure 2 we show detailed results when varying the recurrent computation steps, reporting the mean and standard deviations amongst the seeds.

Irie et al. [18] shown that a recurrent LSTM does well on the Algorithm task with an error rate of 1%. Our experiment shows an LSTM also can solve the Random Walk task. The powerful

---

[2]https://github.com/facebookresearch/transformer-sequential/

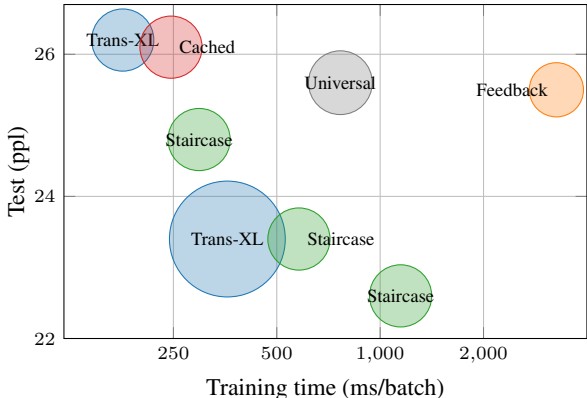

Figure 3: Trade-off between training time and performance (lower ppl is better) on Pushshift.io Reddit. The circle radius is proportional to the number of parameters. The Staircase models can improve performance for longer training time without increasing the model size.

Table 2: **Detailed performance on Pushshift.io Reddit.** We compare our models with varying recurrent steps to baselines with the same number of parameters, a twice as deep (2x) baseline, and Universal Transformers with either the same layer size or a much larger layer size, but same total parameters.

| Model | Num. of params | Recurrent steps | Step size | Forward size | Valid. (ppl) | Test (ppl) | Train batch time (ms) |
|---|---|---|---|---|---|---|---|
| Transformer-XL [9] | 117M | - | - | - | 26.5 | 26.2 | 178 |
| Transformer-XL 2x deep | 218M | - | - | - | 23.7 | 23.4 | 359 |
| Feedback Transformer [13] | 102M | 1 | 512 | 1 | 25.8 | 25.5 | 3260 |
| Cached Staircase | 117M | 1 | 256 | 128 | 26.4 | 26.1 | 246 |
| Staircase | 117M | 2 | 256 | 128 | 25.0 | 24.8 | 297 |
| Universal Transformer | 29M | 2 | - | - | 39.9 | 39.5 | 51 |
| Staircase | 117M | 4 | 256 | 64 | 23.7 | 23.4 | 580 |
| Universal Transformer | 29M | 4 | - | - | 34.9 | 34.5 | 88 |
| Staircase | 117M | 8 | 256 | 32 | 22.9 | 22.6 | 1147 |
| Universal Transformer | 29M | 8 | - | - | 32.3 | 32.0 | 163 |
| Universal Transformer | 120M | 8 | - | - | 25.9 | 25.6 | 766 |
| Universal Transformer | 29M | 16 | - | - | 31.5 | 31.1 | 328 |

Transformer-XL baseline performs poorly here due to its lack of a recurrent mechanism, confirming the results from Fan et al. [13]. The self-attention can access a hidden state far away in the past, but updating that hidden state with a new piece of information brings it up one layer higher. Thus, in a Transformer with $L$ layers, a particular hidden state can be updated only $L$ times before it reaches the final output layer, and becomes unavailable for future computations. This limited computation depth is a problem in the Random Walk task, for example, because the model needs to internally store the agent's location and update it with actions taken at every time step for hundreds of steps.

The Staircase model successfully solves both tasks, even with only two recurrent steps. Thanks to its recurrence through time, the computation depth is only restricted by the input sequence length itself. More concretely, each recurrent step can update the output from the previous step and feed it to the next step, making it possible to maintain and update an internal state without limit.

The Cached Staircase model also performs reasonably well on those tasks. While we only ran this model with $M = 1$ computation step, it is still recurrent in time which is more critical for these tasks than increased computation.

Table 3: Staircase model performance on Pushshift.io Reddit with different forward sizes. The step size is also changed to keep the recurrent steps constant for each section.

| Model | Recurrent steps | Step size | Forward size | Valid. (ppl) | Batch time (ms) |
|---|---|---|---|---|---|
| Cache St. | 1 | 288 | 32 | 26.4 | 489 |
| Cache St. | 1 | 320 | 64 | 26.4 | 318 |
| Cache St. | 1 | 384 | 128 | 26.4 | 246 |
| Staircase | 2 | 128 | 64 | 25.2 | 310 |
| Staircase | 2 | 256 | 128 | 25.0 | 297 |
| Staircase | 4 | 128 | 32 | 23.8 | 605 |
| Staircase | 4 | 256 | 64 | 23.7 | 580 |

The Feedback Transformer solves both tasks, which is not surprising as it is a particular case of a Cached Staircase model with a forward step $C = 1$. However, such fine-grained steps make it slow to train in practice because of the reduced parallelism, as we will see in the analysis in the next section.

### 4.2.2 Staircase models outperform Transformers for the same number of parameters on language modeling tasks

Table 1 shows results on the three language modeling tasks, Enwik8, Pushshift.io Reddit and BASE Data. We show performance versus recurrence plots for the first two tasks in particular in Figure 2. We also show more detailed performance numbers on the Pushshift.io Reddit task in Table 2. In all three tasks, one general trend is that more recurrent steps improve the performance significantly. On the Pushshift.io Reddit task, we see a ~4 perplexity point improvement over the Transformer-XL baseline without adding any new parameters when using 8 recurrent steps, and a ~5 point improvement on BASE Data. Making a twice as deep Transformer-XL (marked with "2x" in Table 2) improves the baseline at the expense of having twice as many parameters than the Staircase models, but is still ~1 perplexity point worse, showing the power of our recurrent models. Our models provides a new way of improving model performance without increasing the number of parameters that is generally applicable.

On Enwik8, we saw a smaller improvement with Staircase model over a Transformer-XL. This could be due to the long context requirement of the character-level data of Enwik8. The Staircase model tries to compress past context into a fixed number of hidden states, equal to $NC - C$ backward tokens to be precise. The Cached-Staircase model works better on this dataset, and only 0.1bpc behind the Feedback Transformer despite being much faster to train. We also trained an LSTM model of a similar size on Enwik8 as a baseline, but its performance was far worse at 1.38 test bpc, showing a simple recurrence alone is not sufficient.

Table 2 also shows the time it takes for training on a single batch for each model. Models with more recurrent steps take longer to run as they have to perform more computations per token, but are still tractable and much faster than the Feedback Transformer, as can be seen in Figure 3. The Feedback Transformer does not perform more computations, but it is slow because it processes one token at a time and also generally performs worse in our language model experiments. In contrast, the Staircase model is fast because it processes $NC$ tokens in parallel despite being recurrent in time. While the Cached Staircase model did not bring much performance improvement on this task, we can also see it does not increase the training time, and that is because it does not add any new computation when $M = 1$. The memory usage during training increases with more recurrent steps, but inference time memory usage will stay the same because we only need keep the latest layer computation in memory.

### 4.2.3 Staircase model's forward size and step size control its performance

The forward step chunk size $C$ and overall staircase step size $NC$ are hyperparameters in Staircase models, where the effective number of recurrent steps is determined by those choices of $N$ in the Staircase model, or truncated to only $M$ steps due to caching in the Cached Staircase model. In Table 3, we compare different values of step size and forward size on the Pushshift.io Reddit task for those models with differing numbers of recurrent steps. We see that, in general, the models are robust

to different choices of those values. Larger forward step sizes are preferable in terms of computational efficiency because they allow more parallelism, but if they are too large some performance in terms of perplexity is lost. We see evidence of this in Table 6 and 7 in Appendix B where both Staircase and Cached Staircase models perform poorly as its forward size $C$ increases.

## 5   Conclusion

In this work, we proposed Staircase attention, which re-introduce recurrence back into the family of Transformer-based models across both time and depth. We show that our Staircase model is able to solve tasks which require tracking of state that conventional Transformers cannot via its recurrence in time. It also delivers more *modeling power per parameter* than conventional Transformers via its recurrence in depth, thus also giving improved performance in language modeling tasks for the same number of parameters, which is especially important in regimes which are memory rather than compute bound. The Cached Staircase variant trades off depth-recurrency for efficiency, but still maintains time-recurrency and do well on the state-tracking tasks as well as a character-level language modeling task. Future work should continue to investigate how recurrence can be built into sequence models, as without a memory component our systems will always be limited to only short-term reactive tasks with limited input. The approaches detailed here are one way forward and should be studied in further applications.

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
