# A Task Setups

Table 4: Shared hyperparameters for all models, given for each task.

| Hyperparameter | Random Walk | Algorithm | Reddit/BASE | Enwik8 |
|---|---|---|---|---|
| Layers | 4 | 4 | 8 | 8 |
| Hidden size | 256 | 256 | 512 | 512 |
| Head count | 4 | 4 | 8 | 8 |
| Dropout rate | 0.2 | 0.2 | 0.3 | 0.3 |
| Embed. dropout | - | - | 0.2 | 0.2 |
| BPTT (i.e. segment) len | 128 | 128 | 256 | 256 |
| Batch size | 512 | 256 | 512 | 512 |
| Learning rate (LR) | 1e-4 | 1e-4 | 7e-4 | 7e-4 |
| Gradient clip | 0.1 | 0.1 | 0.1 | 0.1 |
| LR warm-up steps | 1k | 1k | 8k | 8k |

We provide the hyperparameter setups shared across our models for each task in Table 4. In addition, the hyperparameters tuned for each model for the best performance are shown in Table 5, which were selected using validation performance. We also provide a textual description of some aspects of the base models below.

**Random Walk** We train 4-layer models with a hidden size of 256 and 4 attention heads. We use a learning rate of 1e-4 and 1000 warmup updates to train the models. They are trained for 50k updates with batch size 512. The global staircase models are trained for 400k updates since they need longer to converge. We ran each setting 10 times, except for the Cached Staircase model which was run 5 times.

**Algorithm** We train the 4-layer model with a hidden size of 256 and 4 attention heads. Models are trained to 100k updates with batch size of 256 and learning rate of 1e-4, 1000 warmup updates. We train the global staircase models for 400k steps. We ran each setting 10 times, except for the Cached Staircase model which was run 5 times.

**Pushshift.io Reddit** We train 8-layer models with hidden size of 1024, 8 attention heads. They are trained for 100k updates with a learning rate of 7e-4, 8000 warmup updates and a batch size of 512.

**BASE Data** We train 8-layer models with hidden size of 1024, 8 attention heads. They are trained for 80k updates with a learning rate of 7e-4, 8000 warmup updates and a batch size of 512.

**Enwik8** We train 8-layer models with 8 attention heads. They are trained for 100k updates with a learning rate of 7e-4, 8000 warmup updates and a batch size of 512.

# B Further Detailed Results

Detailed results for a number of our tasks beyond those results reported in the main paper are provided in Tables 6, 7, 8 and 9.

## B.1 Global Cached Staircase Model

For sequence lengths that are not excessively long, it may be desirable at any stage of computation to always have access to all the tokens from the past, whereas the models discussed so far have the limit of $NC$ tokens, see Figures 1b and 1c. We can extend the Cached Staircase model to look back across all tokens, called the Global Cached Staircase. This is achieved by increasing $N$ by one with each step, so *all* prior representations $\mathcal{H}_i^M$ are in the cache and available during later computations. We still employ the cache hyperparameter $M$ as before to control the amount of recurrence and computation necessary during the steps of processing.

However, this Global Cached Staircase model did not perform any better on the state-tracking tasks, see Tables 6 and 7, so we do not consider them in further experiments.

Table 5: Hyperparameters for best performing models across all tasks.

| Tasks | Models | Recurrent steps | Step size | Forward size | Attention span ($S$) |
|---|---|---|---|---|---|
| Random Walk | Staircase | 8 | 64 | 8 | - |
| | Cached staircase | 1 | 256 | 4 | - |
| Algorithm | Staircase | 8 | 64 | 8 | - |
| | Cached staircase | 1 | 64 | 4 | - |
| Reddit | Staircase | 8 | 256 | 32 | - |
| | Cached staircase | 1 | 384 | 128 | - |
| BASE Data | Staircase | 8 | 256 | 32 | - |
| | Cached staircase | 1 | 384 | 128 | - |
| Enwik8 | Staircase | 4 | 256 | 64 | - |
| | Cached staircase | 1 | 260 | 4 | - |

Table 6: Algorithm task detailed results.

| Models | Recurrent steps | Step size | Forward size | Valid (err. %) | Test (err. %) |
|---|---|---|---|---|---|
| Transformer-XL | - | - | - | $59.1 \pm 12.5$ | $59.1 \pm 12.4$ |
| Feedback Trans. | 1 | 32 | 1 | $0.3 \pm 0.0$ | $0.3 \pm 0.0$ |
| Staircase | 8 | 64 | 8 | $12.8 \pm 16.5$ | $12.6 \pm 16.2$ |
| Staircase | 4 | 64 | 16 | $0.5 \pm 0.5$ | $0.5 \pm 0.7$ |
| Staircase | 2 | 64 | 32 | $0.6 \pm 0.2$ | $0.5 \pm 0.2$ |
| Staircase | 2 | 128 | 64 | $48.5 \pm 48.1$ | $48.5 \pm 48.1$ |
| Cached Staircase | 1 | 64 | 4 | $24.6 \pm 21.8$ | $24.3 \pm 21.7$ |
| Cached Staircase | 1 | 64 | 8 | $31.7 \pm 28.7$ | $31.3 \pm 28.7$ |
| Cached Staircase | 1 | 64 | 16 | $27.8 \pm 13.6$ | $27.3 \pm 13.6$ |
| Global Cached Staircase | 1 | 512 | 8 | $0.0 \pm 0.1$ | $0.0 \pm 0.1$ |
| Global Cached Staircase | 1 | 512 | 16 | $7.1 \pm 19.3$ | $7.1 \pm 19.3$ |
| Global Cached Staircase | 1 | 512 | 32 | $20.0 \pm 23.0$ | $19.7 \pm 22.7$ |

## C   Computational Resources

All experiments were run in an internal cluster using 32GB V100 GPUs.The usage varies on recurrent steps; here, we list the maximum resources used in experiments.

- Random Walk experiment used maximum 8 GPUs for ~7 hours.
- Algorithm experiment used maximum 2 GPUs for ~30 hours.
- Language modeling experiments used maximum 32 GPUs for ~30 hours. Experiments were run only once.

Table 7: Random Walk task detailed results.

| Models | Recurrent steps | Step size | Forward size | Valid (%) | Test (%) |
|---|---|---|---|---|---|
| Transformer-XL | 1 | - | - | $90.1 \pm 4.6$ | $90.1 \pm 4.6$ |
| Feedback Trans. | 1 | 64 | 1 | $0.1 \pm 0.0$ | $0.1 \pm 0.0$ |
| Staircase | 8 | 64 | 8 | $0.2 \pm 0.1$ | $0.2 \pm 0.1$ |
| Staircase | 4 | 64 | 16 | $0.2 \pm 0.2$ | $0.2 \pm 0.2$ |
| Staircase | 2 | 64 | 32 | $1.0 \pm 1.3$ | $1.0 \pm 1.2$ |
| Staircase | 2 | 128 | 64 | $42.1 \pm 48.1$ | $42.1 \pm 48.1$ |
| Cached Staircase | 1 | 256 | 4 | $0.1 \pm 0.0$ | $0.1 \pm 0.0$ |
| Cached Staircase | 1 | 256 | 8 | $1.9 \pm 2.0$ | $1.9 \pm 2.0$ |
| Cached Staircase | 1 | 256 | 16 | $27.2 \pm 8.0$ | $27.3 \pm 8.2$ |
| Global Cached Staircase | 1 | 512 | 8 | $0.0 \pm 0.0$ | $0.0 \pm 0.0$ |
| Global Cached Staircase | 1 | 512 | 16 | $1.4 \pm 0.6$ | $1.3 \pm 0.5$ |
| Global Cached Staircase | 1 | 512 | 32 | $52.4 \pm 16.4$ | $52.4 \pm 16.4$ |

Table 8: Results on pushshift.io Reddit with Episodic data. Here, we perform experiments where we prepare an episodic version of the data, where we keep the text length fixed to 256 BPE tokens. The shorter episodes are padded, and longer ones are split into two.

| Model | Recurrent steps | Step size | Forward size | Valid. (ppl) | Test (ppl) |
|---|---|---|---|---|---|
| Transformer-XL | - | - | 256 | 27.6 | 27.3 |
| Cached Staircase | 1 | 256 | 32 | 27.9 | 27.6 |
| Cached Staircase | 1 | 256 | 64 | 27.8 | 27.6 |
| Cached Staircase | 1 | 256 | 128 | 27.6 | 27.3 |
| Staircase | 2 | 256 | 128 | 26.7 | 26.4 |
| Staircase | 4 | 256 | 64 | 25.2 | 24.9 |
| Staircase | 8 | 256 | 32 | 24.3 | 24.0 |

Table 9: Enwik8 task detailed results.

| Models | Recurrent steps | Step size | Forward size | Valid (ppl) | Test (ppl) |
|---|---|---|---|---|---|
| Transformer-XL | - | 256 | 256 | 1.17 | 1.15 |
| Feedback Trans. | 1 | 256 | 1 | 1.13 | 1.12 |
| Cached Staircase | 1 | 260 | 4 | 1.14 | 1.13 |
| Cached Staircase | 1 | 288 | 32 | 1.15 | 1.13 |
| Cached Staircase | 1 | 320 | 64 | 1.15 | 1.13 |
| Cached Staircase | 1 | 384 | 128 | 1.15 | 1.13 |
| Staircase | 2 | 256 | 128 | 1.15 | 1.14 |
| Staircase | 4 | 256 | 64 | 1.14 | 1.14 |