# OpenReview forum: "Staircase Attention for Recurrent Processing of Sequences"
_NeurIPS.cc/2022/Conference — NeurIPS 2022 Accept_

### Official Review · Reviewer_FFvF · 2022-07-06

**Rating:** 7
**Confidence:** 4
**Soundness:** 3 good
**Presentation:** 3 good
**Contribution:** 3 good

**Summary:**

This paper introduces recurrences into transformers with staircase attention. Most existing transformer models process the entire input sequence through multiple layers. Staircase attention departs from this. It segments the input sequence into multiple chunks. As the model processes the input vertically (i.e., feeding into upper layers), it “slides forward” along the input, receiving a new chunk of input while dropping the oldest one. The neural architecture used to process the input chunks is the same as the transformer. To improve the model’s efficiency and flexibility, a cached variant is proposed, disabling the gradient updates of certain (older) input chunks, thereby reducing the computational cost. The proposed models are evaluated on multiple sequence modeling benchmarks, including variants of language modeling tasks and others. Results show that staircase attention achieves a better tradeoff between modeling sizes (in terms of the number of parameters) and performance and that the recurrence it introduces is crucial in certain tasks.

The paper is interesting and novel to the best of my knowledge, and it is clearly written. Most of the claims are reasonable, except that some important experiment details are missing. This is the reason why my initial review is a bit reserved. I’m happy to revise it if the authors can clarify some details (see below in the question section).


**Questions:**

- Could the authors clarify the connections to sliding-window attention?
- For the “random walk” and “algorithm” tasks, which representation is fed into the classification layer?
- Have the authors compared to a staircase baseline ablating the recurrences? For example, such a baseline for a “4-layer staircase transformer with N=8” would be, a canonical 32-layer transformer model, but ties the parameters of layers 1–8, 9–16, 17–24, and 25–32. This baseline has more computational overhead, but is important to establish the claims on the benefits of recurrences.
- In the language modeling experiments, say we have a staircase model with C=1. Is it the case that the model always has the access to N-1 preceding tokens when making a prediction? This relates to the “context size” in [7], and has a substantial impact on transformer language models’ performance. It would be great if the authors can clarify and make sure the models are comparable.

[7] https://arxiv.org/abs/1809.10853

**Limitations:**

The paper did not explicitly discuss the limitations or the potential negative societal impact.

**Strengths And Weaknesses:**

Strengths:
- Staircase attention is interesting and novel to my knowledge.
- The experiments cover a diverse set of sequence modeling tasks, and have a detailed analysis.
- Well-written.

Weaknesses:
- Details on the context sizes need to be clarified (see below)
- The claim on the benefits of recurrences could be better supported (see below).
- Missing discussion of several recent recurrences <-> transformer works [1,2,3,4,5,6].
- Efficiency discussion should be complemented by a discussion on memory overhead, time overhead of the entire training procedure, inference efficiency, etc.

[1] https://arxiv.org/abs/2006.16236
[2] https://arxiv.org/abs/2103.02143
[3] https://arxiv.org/abs/2102.11174
[4] https://arxiv.org/abs/2106.06295
[5] https://aclanthology.org/2021.emnlp-main.602/
[6] https://arxiv.org/abs/2110.02488

---

> ### Author Response · Authors · 2022-08-02
> **Response to reviewer FFvF**
>
> We thank the reviewer for the useful review.
>
> **“Could the authors clarify the connections to sliding-window attention?”**
>
> Our baseline Transformer-XL uses sliding-window attention because each token attends to a fixed-number of previous tokens. The Staircase model has a similar sliding attention window, but its unit is a chunk of C tokens instead of one. In particular, each chunk attends to previous N-1 chunks. Thus, we can view the Staircase as a sliding-window attention, but at chunk level. We will add this discussion to the camera-ready version.
>
>
> **“For the “random walk” and “algorithm” tasks, which representation is fed into the classification layer?”**
>
> Those tasks require prediction at every time step like other language modeling tasks, so we use the same training routine as language modeling. At every time step, a hidden state vector of the last layer is fed to a classification layer, which is trained to predict labels given at every time step.
>
>
> **“Have the authors compared to a staircase baseline ablating the recurrences?”**
>
> We compare to standard Transformers, which do not have recurrences, and to variants with shared layers also without recurrences, namely various versions of Universal Transformers. We find improvements in several settings.
>
> **“In the language modeling experiments, say we have a staircase model with C=1. Is it the case that the model always has the access to N-1 preceding tokens when making a prediction? This relates to the “context size” in [7], and has a substantial impact on transformer language models’ performance. It would be great if the authors can clarify and make sure the models are comparable.”**
>
> Yes, each token will be attending to previous N-1 tokens, which we call “attention-span”. We made sure that the baselines have a comparable attention-span as our models. However, the notion of context length is more complicated to define because multiple attention layers are stacked. In Transformer-XL, for example, a token can attend to previous L steps, but those tokens also attends to their previous L steps. So the effective context size would be L times the number of layers. See Transformer-XL paper (https://arxiv.org/abs/1901.02860) for more details on its context size. In contrast, the Staircase stacks attention layers infinitely as long as input sequence continuous. This means thee theoretical limit of the context size would be infinite similar to recurrent neural nets.
>
>
> **“Missing discussion of several recent recurrences ↔ transformer works [1,2,3,4,5,6]”**
>
> Thank you for pointing out additional related work, we will incorporate them in the related works section of our camera-ready versions.
>
>
> **“Efficiency discussion should be complemented by a discussion on memory overhead, time overhead of the entire training procedure, inference efficiency, etc.”**
> There is no memory or computation overhead during training and inference. This is because Staircase model functions very similar to Transformers. With increased recurrence steps, training time per batch do increase as shown in Table 2. The memory usage during training would also increase, but inference time memory usage will stay the same because we only need keep the latest layer computation in memory. However, this also improves the performance. We will add this discussion to the paper.

---

> > ### Comment · Reviewer_FFvF · 2022-08-03
> > **Thanks for the response**
> >
> > Most of my concerns have been addressed, I have updated my review accordingly.

---

### Official Review · Reviewer_Cujd · 2022-07-08

**Rating:** 7
**Confidence:** 5
**Soundness:** 4 excellent
**Presentation:** 3 good
**Contribution:** 4 excellent

**Summary:**

The paper builds upon the Feedback Transformer (Fan et al., 2020) to introduce recurrence into the Transformer architecture. While a regular transformer processes C tokens in parallel, the proposed approach - staircase attention - processes C new tokens at each time step and NC-C tokens from prior time steps. Thus, the transformer core moves ahead in time and resembles a staircase. The output for a given token is generated after N passes. The approach is compute-heavy compared to a regular transformer but uses the same number of parameters. To save computation, the paper proposes a variant that caches states to decrease recurrence while keeping the context size fixed. The paper shows that staircase attention outperforms transformer variants (transformer XL) on state tracking tasks (Random walk and Algorithm) as well as language modeling (Enwik8, Reddit, BASE). The staircase attention obtains a similar performance as the Feedback transformer on the state tracking tasks and improves upon it on some language models tasks (Reddit/BASE).



**Questions:**

* Although the cached version of the staircase attention is similar to the feedback transformer, they do not perform similarly on Algorithm and BASE. It would be useful to provide more analysis to understand why this is the case.
* Why does validation error increase when going from 6 to 8 recurrence steps for the 'Algorithm' task?
* Enwik8: "The Cached-Staircase model works better on this dataset, and only 0.1bpc behind the Feedback Transformer despite being much faster to train. ". It'd be good to understand why Cached-Staircase performs better than standard staircase for this task.
* Eqn 2: Please mention that TC stands for 'transformer core' when this acronym is used for the first time.

Typos:
* Figure 1: 'indentity' -> 'identity'
* Sec 2: 'non-recurrent models have superseded in popularity recurrent' -> 'non-recurrent models have superseded recurrent' or 'non-recurrent models have become more popular than recurrent'



**Limitations:**

Yes, the authors have addressed limitations. They have not discussed potential negative societal impact of their work.

**Strengths And Weaknesses:**

Strengths:
* Novel idea to introduce recurrence into a transformer architecture
* Architecture allows a control between amount of recurrence and context size
* Proposes a cached version which controls the amount of recurrence and provides speeds at the cost of performance
* Demonstrates improvements over Feedback transformer in terms of training speed

Weaknesses:
* It would be useful to add results from standard transformer and LSTM for all state-tracking/LM tasks in Table 1. Such a comparison would be useful to all researchers.
* It would be useful to provide some more analysis (see below).
* Given that staircase attention is compute heavy, it would be useful to see a comparison of the inference speed between all models for say one of the tasks (e.g. Reddit). This would enable researchers to understand the trade-off between performance gains vs added compute.

---

> ### Author Response · Authors · 2022-08-02
> **Response to reviewer Cujd**
>
> Thank you for the insightful review and suggestions. Here are our responses to the questions:
>
>
> **“It would be useful to add results from standard transformer and LSTM for all state-tracking/LM tasks in Table 1.”**
>
> We do have LSTM results on those tasks mentioned in the main text, but we will add those numbers to Table 1 for clarity. Our Transformer-XL baseline is an almost standard Transformer, but with just few modifications (caching of previous hidden states and relative position embedding) to make it more efficient at processing long sequences in our language modeling tasks. Applying a vanilla Transformer without those changes to long-context language modeling is extremely inefficient, especially during inference time because a whole block of tokens need to be processed to predict just one token. See the Transformer-XL paper (https://arxiv.org/abs/1901.02860) for more details.
>
>
> **“Given that staircase attention is compute heavy, it would be useful to see a comparison of the inference speed between all models for say one of the tasks (e.g. Reddit). This would enable researchers to understand the trade-off between performance gains vs added compute.”**
>
> In Table 2, we provide processing time for a single batch for all models on the Reddit task. This clearly shows the trade-off between compute efficiency and performance. More recurrent steps improve the performance, but also costs more train time. However, our best performing model is still much faster to train than the Feedback baseline. While this time include both forward and backward passes, we expect the forward-only inference time to have a similar trend as two operations have similar operations.
>
>
> **“Although the cached version of the staircase attention is similar to the feedback transformer, they do not perform similarly on Algorithm and BASE. It would be useful to provide more analysis to understand why this is the case.”**
>
> Their difference on Algorithmic task is only 1%, which probably can be attributed to training noise. But on BASE, yes the Feedback is performing better than the Cached Staircase. This is because the Cached Staircase has less frequent recurrency than Feedback. In Feedback, each token is recurrent so it can attend to the LAST hidden state of the previous token, which is crucial for its performance boost. In contrast, the Cached Staircase is recurrent at every C tokens, so the last hidden state is not accessible to nearby C tokens. In table 9 of the supplementary material, we can see the Cached Staircase’s performance improve as C decrease, and will actually match the Feedback at C=1 because they are identical.
>
>
> *“Why does validation error increase when going from 6 to 8 recurrence steps for the 'Algorithm' task?”*
> A likely reason here is that more recurrent steps means more non-linear layers between input and output. Deeper Transformer networks are known to be more difficult to optimize (see https://aclanthology.org/D18-1338.pdf). With 8 recurrent steps, there are effectively 32 layers and perhaps this is why some seeds failed to learn this task.
>
> **“Enwik8: "The Cached-Staircase model works better on this dataset, and only 0.1bpc behind the Feedback Transformer despite being much faster to train. ". It'd be good to understand why Cached-Staircase performs better than standard staircase for this task."**
>
> We have more detailed Enwik8 results in Table 9 of the supplementary material. There we can see that the Staircase performs similarly to the Cached Staircase when the forward step size is matched. Since the Cached Staircase is efficient, we were able to reduce the forward step for better performance, but doing the same for the Staircase is computationally challenging. Enwik8 is a character-level data requiring a long context, so it is undesirable to reduce the steps size. When the step size is constant, reducing the forward step will lead to more recurrent steps, which will requires more computation.

---

> > ### Comment · Reviewer_Cujd · 2022-08-09
> > **Satisfied with the response**
> >
> > Thanks for the authors for their willingness to update the paper with additional results and the explanations for the above questions.

---

### Official Review · Reviewer_3i3K · 2022-07-10

**Rating:** 6
**Confidence:** 5
**Soundness:** 3 good
**Presentation:** 3 good
**Contribution:** 2 fair

**Summary:**

The paper falls into a set of efforts that incorporate recurrence mechanisms in Transformer-based models while keeping the model reasonably efficient. While Transformers have been outperforming recurrent models in a number of practical tasks, some prior works have highlighted limitations of Transformer's feedforward-like architecture on tasks that involve state-tracking (among others).

Several prior works have tried to integrate recurrence in a Transformer-like architecture to mitigate such limitations but most of them end up being computationally inefficient compared to standard Transformers. One such approach was Feedback Transformer (FeedT) which this paper builds on. Although feedback Transformer reportedly performed better than Transformer across different tasks, it suffered from efficiency issues. This paper takes a step towards mitigating the efficiency issues while preserving some of the benefits of recurrence.

They propose a modified version of attention architecture namely Staircase attention which recurrently processes different chunks/blocks of tokens and processes each chunk in a parallel manner. The computation for each chunk depends on the state of some previous chunks which makes the model recurrent. Compared to FeedT which incrementally processes each token (like an RNN) this reduces the computational cost by a factor of the size of the chunks. Additionally, each chunk can go through multiple recurrent computations. They also introduce a cached version of the model where the number of recurrent computations for each chunk is limited to reduce computational costs. They compare their approach against several Transformer variants and report the performance on 3 language modeling and 2 synthetic algorithmic tasks. Their results indicate that Staircase models achieve near-perfect accuracy in synthetic tasks while performing well on language modeling tasks.

**Questions:**

(Q1) Is there any limitation of processing chunks compared to the standard recurrent model? Maybe when the chunk size is large? I do not think there would be any limitation in terms of expressiveness of the model until recurrence is removed but there could be differences in performance as the chunk size grows larger.


(Q2) Table 2: Are the recurrent steps in universal transformers and the Staircase model comparable?

(Q3) Is there any relation between the proposed model to the Block-recurrent Transformers method [1]? On a high level, there seems to be some similarity in terms of processing blocks of tokens in parallel and recurrently processing blocks.


[1] BLOCK-RECURRENT TRANSFORMERS. https://arxiv.org/pdf/2203.07852.pdf



Typos:

Line 55
Figure 1: Indentity -> Identity?

**Limitations:**

Not clear where limitations are explicitly discussed in the paper.

**Strengths And Weaknesses:**



### Strengths

(S1) **Simplicity.** The proposed idea seems simple and does not seem to make tedious modifications to standard Transformers. From the results, it seems to achieve the advantages of both recurrence (strong performance on state tracking tasks) and attention (in terms of preserving efficiency) while obtaining competitive performance on practical language modeling tasks.

(S2) **Good results.** The experimental setup and evaluation seem sound. In particular, relevant baselines such as universal Transformers and Transformers-XL have been included apart from direct ones such as feedback Transformers. The results indicating slightly better performance on LM tasks with a similar number of parameters and training time look promising.


(S3) **Writing.** The paper is well written and the general narrative of the paper is easy to follow. The problem is well-motivated and the experiments/results are well-explained. However, there is some lack of clarity about the details of the model (more on that below).



### Weakness



(W1) **Evaluation.** The set of tasks seems quite narrow and it seems like a lot more could have been done to support the efficacy of the proposed approach. The method is evaluated on 2 synthetic tasks and 3 LM tasks. Although the results on those tasks look promising it does not seem to provide a broad picture of the effectiveness of the method. The improvements in LM tasks are also incremental which is not an issue if the method consistently performs competitively across more tasks. I believe a comprehensive analysis across different tasks and across different aspects of the model is important to judge its effectiveness.

**[Possible Improvements]**

Tasks. Since the method builds on feedback Transformers I was expecting most tasks explored in that paper to be naturally a part of the experiments here. For instance, copy, reverse, and counting should be easy to implement and experiment with. There are formal language tasks where standard Transformers have also been found to struggle compared to RNNs. If the method can perform well on those while maintaining efficiency it could strengthen the case. On the practical side, long document datasets such as GitHub, arxiv, and PG19 datasets could be explored. Positive results on some of the above-mentioned tasks could significantly help strengthen the case for the proposed method.

Model Analysis. The effect of varying parameters such as chunk size and recurrence across depth is unclear. For algorithmic tasks, it could be easier to study the effect of increasing the chunk size. I would guess as the chunk size becomes very large, the model would get closer to standard Transformers and its performance might decrease. Similarly, it could be helpful to highlight if and where recurrence across depth leads to any advantage.

(W2) **Novelty.** The contribution in terms of algorithmic novelty is weak in my opinion. Given the previous approaches, it seems like an extension to reduce computational costs. The recurrence across depth seems different but it is not clear if it leads to improvements. In my opinion, it is not necessarily an issue if the method consistently achieves strong performance across several tasks.


Other Comments.

(W3) **Method description.** The main method (Staircase attention model) is not very clearly described. It seems to be described in an informal and conceptual manner. Although the method is clear up to a certain degree, I am not confident if I were to reimplement it based on my understanding then it would be the same as the author's version.
[Suggestion] It would be helpful to the readers if a very detailed description of the model is provided in the appendix.

---

> ### Author Response · Authors · 2022-08-02
> **Response to Reviewer 3i3K**
>
> Thank you for the very detailed review and recommendations. However, we would like to address few points raised as weakness:
>
> **“Tasks seems quite narrow”**
> We respectfully disagree. We experimented on 5 different tasks ranging from algorithmic tasks to real-world large-scale language modeling. The two toy tasks are complex enough that powerful Transformer models fail to solve them. They clearly demonstrate the advantage of the time-recurrency of the Staircase model. We also included a character-level language modeling task Enwik8 that requires much longer context than word-level tasks. In addition, we experimented on an extremely large-scale dataset BASE that consists of 100B tokens (PG19 has only 2B tokens in comparison) and even contains whole books. The experiments on this dataset shows that our model can scale and work well on long-context tasks. While it is possible to keep adding more tasks using more GPUs hours, we consider the current set of tasks sufficient for demonstrating the advantage of the proposed model.
>
> **“The effect of varying chunk size and recurrence across depth”**
> We do have experiments showing this. Figure 2 shows the effect of varying the recurrent steps on 4 different tasks. Table 2 have more detailed analysis that includes training efficiency. In Table 3, we vary the chunk size for different recurrent steps. In the supplementary material, we have more detailed results on varying the chunk size and the recurrent steps, and its effect on performance. In addition, we added new experiments (see our answer to the first question below) with larger chunk sizes to observe this behavior on the toy tasks that require recurrency. From those experiments, we observe that more recurrence across depth lead to better performance on word-level language modeling tasks. Also, on the state-tracking tasks, too large chunk-size causes performance drop due to its less frequent recurrency.
>
> **“The contribution in terms of algorithmic novelty is weak ... like an extension to reduce computational costs. ”**
> We proposed a completely new way of achieving time-recurrency through the attention mechanism. Yes, the model reduces the computational cost of such a recurrent architecture, which is also an important contribution, especially for Transformer-based models that require a lot of compute.
>
> **“Although the method is clear ... same as the author's version.”**
> We will release our code so it would be easier for other people to use it. This will also help people to understand the exact implementation details and enable them to reimplement. We will also add a section in the appendix about exact model implementations in the camera-ready version.
>
> Below are our answers to the questions asked:
>
> **“Is there any limitation of processing chunks? when the chunk size is large?  differences in performance as the chunk size grows larger?”**
> As the chunk size (i.e. forward step size) increase, the recurrency of the model will be less frequent in time, eventually converging to a normal Transformer. We added new experiments with larger chunk sizes to observe this behavior on the toy tasks that require recurrency.  Both at recurrent step 2, a staircase of large step size (128), forward size 64 preforms worse in both algorithm and random walk tasks than a model with step size 64, forward size 32 . On algorithmic task, it lands at 48.5 ±48.1  on valid and 48.5 ±48.1 on test, comparing to 0.6 ±0.2 on valid and  0.5 ±0.2 on test. On the random walk task, it showed 42.1 ±48.1 on valid, and 42.1±48.1 on test, comparing to 1.0±1.3 (valid), 1.0±1.2 (test). Note this model is still outperforming a transformer-XL baseline thanks to its ‘reduced’ recurrency. We will add these experimental results to the camera-ready version.
>
> **“Table 2:universal transformers and the Staircase model comparable?”**
> Yes, we matched their recurrence in the model depth. Unlike Staircase, Universal transformer lacks recurrence in input sequence direction, but both models are recurrent in their depth by repeating the same operation multiple times. In fact, Universal transformer requires all input tokens to be presented at the same time, so it cannot process stream of data like LSTM or Staircase. We also have a Universal transformer with a larger hidden dimension so it has the same number of parameters (120M) as the Staircase.
>
> **“Relation with Block-recurrent Transformers”**
> Block-recurrent Transformer is introduced recently as a way of adding recurrency to a Transformer. Like the Staircase, it also adds recurrency in the input sequence direction. However, the recurrency mechanisms are very different. In the Staircase, the recurrency is intermediated by existing the existing self-attention mechanism. In contrast, the Block-recurrent Transformer essentially adds a new Transformer model for maintaining recurrent states. This new Transformer runs orthogonal to the existing self-attention, and the two models exchange information via cross-attention layers.

---

### Official Review · Reviewer_ZfSh · 2022-07-12

**Rating:** 6
**Confidence:** 3
**Soundness:** 3 good
**Presentation:** 3 good
**Contribution:** 2 fair

**Summary:**

In this paper, authors propose a new attention algorithm to operate the input sequence recurrently. Each time, the staircase attention process the seen and unseen tokens at the same time. They conduct several experiments on various tasks including ones which can be done by the standard Transformer. For other tasks, their method try to obtain a better efficiency vs performance tradeoff.

**Questions:**

Please refer to the above section.

**Limitations:**

Yes.

**Strengths And Weaknesses:**

Strengths:
1. Their idea is easy to follow and their motivation is clear.
2. They conduct a list of experiments to show the performance of their models and compare to other models.
3. Comparison to relevant models is sufficient and clear.

Weakness:
1. My major concern is the tradeoff between the standard staircase attention and cased one. It seems that staircase attention is able to obtain much better performance than cached attention but with low efficiency. For the cached one, it can not achieve great performance compared to other models such as feedback transformer. Can you find a better tradeoff between cache and original staircase attention.
2. Will you release the code? It seems not straightforward to implement the proposed method.
3. Is that possible to try various recurrent steps for cached staircase (or do I miss something here)? Also ablation study between M and N for cached staircase attention.

---

> ### Author Response · Authors · 2022-08-02
> **Response to Reviewer ZfSh**
>
> We thank the reviewer for taking time to review our paper.
>
> **"My major concern is the tradeoff between the standard staircase attention and cased one. It seems that staircase attention is able to obtain much better performance than cached attention but with low efficiency. For the cached one, it can not achieve great performance compared to other models such as feedback transformer."**
>
> We consider the flexibility of having a choice between the Staircase and Cached Staircase as an advantage.  Yes, the Staircase is less efficient than the cached version, but still trains much faster than the Feedback Transformer and achieves better performance as shown Tab2. The Cached Staircase is even more efficient, but still matches the Feedback’s performance on state-tracking toy tasks where Transformer-XL fail. Thus, one can choose a Staircase version that is best suited for the task and compute constraint. Furthermore, the Feedback Transformer is actually a special case of Cached Staircase where the forward step size is 1. We can see the performance of Cached Staircase is improving as we reduce the forward step size in Table 9 in the appendix. It will exactly match the Feedback transformer when forward step is 1, but the training time will as slow as the Feedback transformer.
>
>
> **“Will you release the code? It seems not straightforward to implement the proposed method”**
>
> Yes, we will release our code that used in the paper along with scripts for running experiments.
>
>
>
> **“Can you find a better tradeoff between cache and original staircase attention ...Is that possible to try various recurrent steps for cached staircase (or do I miss something here)?”**
>
> The parameter M controls the amount of compute per token. In the paper, we compared M=N, i.e. original Staircase, and M=1, i.e. the Cached Staircase, showing their advantages. In theory, it is possible to set M between 1 and N if more fine-grained control over efficiency is required.

---

### Meta-Review · Area_Chair_SEWP · 2022-08-23

**Recommendation:** Accept
**Confidence:** Certain

**Metareview:**

The paper proposes staircase attention to model recurrence with the Transformer architecture. Reviewers are generally on the acceptance side, as they believe the proposed approach is interesting and achieves good empirical performance.

**Award:**

No

---

### Decision · Program_Chairs · 2022-09-14

Accept